# Application of a time-series deep learning model to predict cardiac dysrhythmias in electronic health records

Aixia Guo[1], Sakima Smith[2], Yosef M. Khan[3], James R. Langabeer II[4], Randi E. Foraker[1,5]*

**1** Institute for Informatics (I2), Washington University School of Medicine, St. Louis, MO, United States of America, **2** Department of Internal Medicine, The Ohio State University, Columbus, OH, United States of America, **3** Health Informatics and Analytics, Centers for Health Metrics and Evaluation, American Heart Association, Dallas, TX, United States of America, **4** School of Biomedical Informatics, Health Science Center at Houston, The University of Texas, Houston, TX, United States of America, **5** Department of Internal Medicine, Washington University School of Medicine, St. Louis, MO, United States of America

* randi.foraker@wustl.edu

**Data Availability Statement:** The data are owned by a third party and the authors do not have permission to share the data. Requesting access to The Guideline Advantage (TGA) data must be done

## Abstract

### Background

Cardiac dysrhythmias (CD) affect millions of Americans in the United States (US), and are associated with considerable morbidity and mortality. New strategies to combat this growing problem are urgently needed.

### Objectives

Predicting CD using electronic health record (EHR) data would allow for earlier diagnosis and treatment of the condition, thus improving overall cardiovascular outcomes. The Guideline Advantage (TGA) is an American Heart Association ambulatory quality clinical data registry of EHR data representing 70 clinics distributed throughout the US, and has been used to monitor outpatient prevention and disease management outcome measures across populations and for longitudinal research on the impact of preventative care.

### Methods

For this study, we represented all time-series cardiovascular health (CVH) measures and the corresponding data collection time points for each patient by numerical embedding vectors. We then employed a deep learning technique–long-short term memory (LSTM) model–to predict CD from the vector of time-series CVH measures by 5-fold cross validation and compared the performance of this model to the results of deep neural networks, logistic regression, random forest, and Naïve Bayes models.

### Results

We demonstrated that the LSTM model outperformed other traditional machine learning models and achieved the best prediction performance as measured by the average area under the receiver operator curve (AUROC): 0.76 for LSTM, 0.71 for deep neural networks,

by contacting the American Heart Association via email qualityresearch@heart.org. The authors did not have any special access privileges that others would not have. The Python code related to the analyses can be found in Github repository: https://github.com/aixiaguo/CD_prediction/blob/master/CD.py.

**Funding:** The author(s) received no specific funding for this work.

**Competing interests:** The authors have declared that no competing interests exist.

0.66 for logistic regression, 0.67 for random forest, and 0.59 for Naïve Bayes. The most influential feature from the LSTM model were blood pressure.

## Conclusions

These findings may be used to prevent CD in the outpatient setting by encouraging appropriate surveillance and management of CVH.

## Introduction

Cardiac dysrhythmia (CD) is a problem in which the heart has an irregular rhythm [1]. It affects millions of Americans in the United States (US) and approximately 25% of Americans older than 40 years develop a CD [2]. Six million people die annually due to sudden cardiac death caused by ventricular tachyarrhythmias (one type of CD) globally [3]. Risk factors which increase the chance of developing a CD include high blood pressure, diabetes and obesity. CD can be managed in the outpatient setting with medications or behavior change (i.e., diet or physical activity) or in the inpatient setting with cardiac procedures such as an ablation or cardioversion which can restore the rhythm back to normal. If diagnosed and managed appropriately, it can effectively reduce the risk of future blood clots (thrombus formation), heart failure and stroke (thromboembolic events) [4].

Electronic health records (EHR) contain longitudinal healthcare information of patients, including diagnoses, procedures, medications, lab tests and imaging data [5], which could be used for discovering the relationships and predicting patterns from data. For example, a study reported that CD was negatively associated with type II diabetes [6]. Atrial fibrillation (AF) is the most common CD, impacting over 6 million Americans, and multiple factors including clinical, genetic and environmental factors were found to have associations with AF [7–9]. For example, a risk model using data from outpatient clinics (Vanderbilt University Medical Center) predicted AF with demographic information, blood pressure, and smoking status [10]. In this analysis, traditional machine learning algorithms such as Naïve Bayes (NB), support vector machines (SVM) and random forest (RF) [11] along with newly developed algorithms [12] were applied to identify AF using EHR data. In the case of ventricular arrhythmias, informative clinical variables such as blood pressure, treadmill exercise time, and body mass index (BMI) predicted among hypertrophic cardiomyopathy patients using some traditional machine learning algorithms, including RF and logistic regression (LR) [13].

Recently, deep learning algorithms have grown in popularity for data-driven prediction models. Such models can effectively learn from experience by capturing features and dependencies in longitudinal data and have achieved great success in bioinformatics and healthcare fields [14–17]. For example, scalable deep learning methods were developed to accurately predict medical events from two academic medical centers' EHR data and achieved high accuracy in prediction tasks [18]. In this paper, we applied a long-short term memory (LSTM) model [19] on time-series EHR data to explore the contribution of modifiable cardiovascular risk factors to the development of CD in the outpatient setting. Central to our analysis was the characterization of cardiovascular health (CVH) and CD outcomes using EHR data from clinics across the US. We evaluated the association between time-series CVH and CD diagnoses, and hypothesized that CD could be predicted using data commonly recorded in the EHR. To our best knowledge, it is also the first time that deep learning algorithms have been applied to predict CD using time-series EHR data.

## Methods

### Ethics statement

All the data were fully anonymized before we accessed them. Our study was approved by the Institutional Review Board at the Washington University School of Medicine in St. Louis. We obtained a written acknowledgement of proprietary rights and non-disclosure and data use agreement from the American Heart Association (The Washington University_NDA_DUA_-CONTRACTID 158065_2019.04.26_K).

Established in 2011, The Guideline Advantage (TGA) was a clinical data registry jointly operated by the American Cancer Society, the American Diabetes Association, and the American Heart Association [20]. The program collects EHR data to track and monitor outpatient prevention and disease management. Briefly, the data collected through TGA from over 70 clinics provide a unique platform for longitudinal research on the impact of preventative care. The program's research strategy is focused on identifying patient-, provider-, and practice-level factors associated with guideline adherence and assessing the effectiveness of quality improvement interventions in increasing guideline adherence. Here we used TGA data to predict the diagnosis of CD among 362,533 unique patients in the data set.

Our data set represented patients seen in the outpatient setting over a 10-year period (2007 to 2016). We defined our study outcome by classifying 19,597 unique ICD-9 and ICD-10 codes to a smaller number of clinically meaningful categories using Clinical Classifications Software (CCS) [21]. After the codes were converted to the appropriate CCS category, we identified 34,511 patients with a diagnosis of CD (single level CCS code = 106). Among them, the majority (55%) were female patients, and 66% of patients were white. If a patient had multiple CD diagnoses in the data set, only the earliest one was considered.

Next, we extracted all measurements of CVH prior to the diagnosis of CD. We utilized measures of CVH as follows: smoking status, body mass index (BMI), blood pressure, hemoglobin A1c, and cholesterol, which were defined and classified by the AHA into three categories: ideal, intermediate, or poor according to **Table 1**. To classify patients as intermediate health or treated-to-goal for selected CVH submetrics (**Table 1**), we converted the drug names to their drug classes by comparing the drug names in our dataset with the Multum drug database [22]. One string match technique–Levenshtein distance algorithm [23]–was applied and we considered the distance between the two matched strings as less than five to be matched and included these in subsequent analyses.

**Table 1. Measures of CVH which are available in the EHR (adapted from: Lloyd-Jones, 2011) [24].**

| | Poor Health | Intermediate Health | Ideal Health |
|---|---|---|---|
| Health Behaviors | | | |
| Smoking status | Yes | Former $\leq$ 12 months | Never or quit > 12 months |
| Body mass index | $\geq$ 30 kg/m$^2$ | 25–29.9 kg/m$^2$ | < 25 kg/m$^2$ |
| Health Factors | | | |
| Total cholesterol | $\geq$ 240 mg/dL | 200–239 mg/dL or treated to goal | < 200 mg/dL |
| Blood pressure | Systolic $\geq$ 140 mm Hg or Diastolic $\geq$ 90 mm Hg | Systolic 120–139 mm Hg or Diastolic 80–89 mm Hg or treated to goal | Systolic < 120 mm Hg / Diastolic < 80 mm Hg |
| Fasting plasma glucose | $\geq$ 126 mg/dL | 100–125 mg/dL or treated to goal | < 100 mg/dL |

We studied patients with CD who had four or more outpatient CVH measures in the data set (n = 5,271). Using the same criteria, we randomly selected 5,784 patients from the dataset who did not have a diagnosis of CD. In sensitivity analyses, we tested the robustness of our strategy by changing the number of outpatient CVH measures from zero to three, respectively. Ultimately, our data set comprised 11,055 patients who had four or more encounters over the 10-year study period.

## Statistical analysis

To prepare the CVH measures for analysis, we combined the submetric with its classification according to **Table 1**. For example, if a patient had a measurement of "ideal" cholesterol, then we combined the submetric and its value as cholesterolideal. The resulting features were mapped to a 32-dimensional vector by word embeddings [25] in our model. The Genism Word2Vec model was configured the hyperparameters as following: size (embedding dimension) as 32, window (the maximum distance between a target word and all words around it) as 5, min_count (the minimum number of words counted when training the model) as 1, sg (the training algorithm) as CBOW (The continues bag of words). The input of Word2Vec model was all above combined measurements of all 11,055 patients. We also added time information for all measurements as time steps. Each feature was associated with a time point which was calculated by the difference in days between the corresponding visit time and the latest measurement time. For example, if the most recent visit date was February 11, 2019, and measurement was conducted on January 11, 2019, then the time point value is: 31. Thus, each individual patient had its own vector to represent their measurements of CVH.

The embedded vectors of patients were the inputs for our long short-term memory (LSTM) model. We applied an LSTM algorithm to investigate the association between time-series CVH measurements and the outcome of CD. We also investigated other machine learning and deep learning algorithms such as DNN [26], LR [27], RF [28] and NB [29] to study the same association between CVH and CD. All of the CVH measurements for each patient were sorted in chronological order. We padded the patients with virtual events as the same length (311) in the form of $[0_1, \ldots 0_k, event\_1, event\_(311\text{-}k)]$ if they had less events than the maximum number of measures (311), where k was the difference of 311 and number of records that patients had.

To investigate the effects by continuous vectors obtained from Word2Vec algorithm, we conducted the same predictions by using categorical variables. These categorical variables were sorted in a time order, and each categorical event concatenated with the same time points (e.g., difference in days between the corresponding visit time and the latest measurement time) were the inputs of the models of LSTM, DNN, RF, LR, and NB. We did the same padding approach as above for patients had less events than 311.

For each model of predictions, we utilized 5-fold cross validation by dividing dataset into 5 folds with each fold serving as a testing dataset and the remaining 4 folds as a training dataset. Criteria of the area under the receiver operator curve (AUROC) and other metrics, i.e., accuracy, sensitivity, precision, f1 score, and specificity were calculated to evaluate the performance of the models.

## LSTM unit

A common LSTM unit is composed of a cell and three gates: input gate, output gate and forget gate. The cell remembers information at each time step and these gates control the flow of information pass on to and forget/discard to the next time step [30]. We illustrated the basic structure of an LSTM unit as in **Fig 1**.

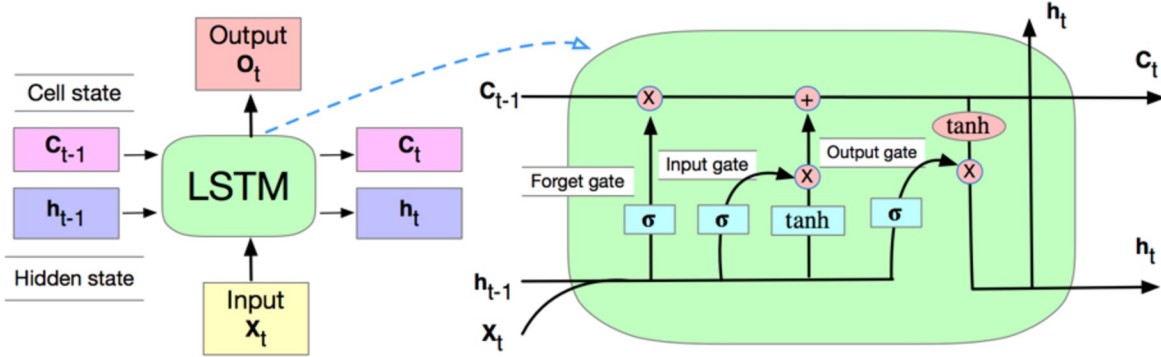

**Fig 1. Graph illustration of LSTM unit.**

Mathematically, the equations for forward pass to update an LSTM unit with a forget gate at a time $t$ are:

$$\text{Forget gate } \mathbf{f}_t = \sigma(\mathbf{W}_f \mathbf{h}_{t-1} + \mathbf{U}_f \mathbf{X}_t + \mathbf{b}_f)$$

$$\text{Input gate } \mathbf{i}_t = \sigma(\mathbf{W}_i \mathbf{h}_{t-1} + \mathbf{U}_i \mathbf{X}_t + \mathbf{b}_i)$$

$$\mathbf{C}_t = \mathbf{f}_t * \mathbf{C}_{t-1} + \mathbf{i}_t * (\tanh(\mathbf{W}_c \mathbf{h}_{t-1} + \mathbf{U}_c \mathbf{X}_t + \mathbf{b}_C)$$

$$\text{Output gate } \mathbf{o}_t = \sigma(\mathbf{W}_o \mathbf{h}_{t-1} + \mathbf{U}_o \mathbf{X}_t + \mathbf{b}_o)$$

$$\mathbf{h}_t = \sigma(\mathbf{o}_t * \tanh(C_t))$$

Where $*$ denotes the element-wise product and $\mathbf{X}_t$ is the input vector (i.e., embedding vector in our case) at time $t$. The weight matrices $\mathbf{W}_f$, $\mathbf{W}_i$, $\mathbf{W}_c$, $\mathbf{W}_o$ for hidden state $\mathbf{h}_t$, $\mathbf{U}_f$, $\mathbf{U}_i$, $\mathbf{U}_c$, $\mathbf{U}_o$ matrices for input $\mathbf{X}_t$, and bias vector parameters $\mathbf{b}_f$, $\mathbf{b}_i$, $\mathbf{b}_c$, $\mathbf{b}_o$ are learned during the training stage and $\mathbf{h}_t$ is the hidden layer output vector. Activation function $\sigma$ is the sigmoid function and *tanh* is the hyperbolic tangent function.

Our LSTM model comprised an input layer, one hidden layer (100 dimensions) and a scalar output layer. A binary cross-entropy loss function was employed as the output layer and a sigmoid function was used as the activation function for the hidden layer. Adam optimizer [31] was used to optimize the model with a mini-batch size of 64 samples. The DNN was comprised of an input layer, 5 hidden layer (with 256, 256, 128, 64 and 32 dimensions respectively). and a scalar output layer. We used the Sigmoid function [32] at the output layer and ReLu function at each hidden layer. Binary cross-entropy was used as loss function and Adam optimizer was used to optimize the models with a mini-batch size of 64 samples. The LR, RF and NB models were configured by default options in the package of Scikit-learn in Python 3.

We then investigated which features were the most important in CD prediction. To obtain this goal, we iterated the model 15 times by setting constant value for one feature each time. For each feature, we first manually set it as a constant (not informative for the predictive models), then tested the prediction performance of trained models using the manually changed features to evaluate the discriminative importance of the given feature. The resulting performance then was compared its prediction accuracy and AUROC with the full model. If there was a large change between these two values, it indicated that this feature was important

and discriminative to the prediction. Analyses were conducted by using the libraries of Scikit-learn, Scipy, Matplotlib with Python, version 3.6.5 in 2019.

## Results

Our study population was 58% female and 53% white (**Table 2**). Approximately 58% of women had been diagnosed with CD and around 60% of CD patients were white. Since patients had multiple encounters, there were multiple measures of CVH. The average number of measures for each patient was 24 and the median was 17.

**Fig 2** displays all the measures and results of two patient examples in which one was diagnosed as CD and the other was not.

**Table 3** lists the numbers of ideal, intermediate and poor measurements for each submetric. As seen in **Table 3**, patients without CD (39%) had a higher prevalence of ideal BMI compared to those with CD diagnoses (23%), and ideal blood pressure measurements followed the same pattern.

Word embeddings produced a vector representation of words which were the features of patients. **Fig 3** shows the embeddings visualization of all of the features projected to the first two components in the t-Distributed Stochastic Neighbor Embedding analysis (tSNE) [33].

**Table 2. Characteristics [mean, (SD) or n (%)] of the study population.**

| | |
|---|---|
| Gender [n (%)] | |
| Female | 6379 (57.7) |
| Male | 4673 (42.3) |
| Other/Unknown | 3 (0.0) |
| Gender with CD | |
| Female | 3054 (57.9) |
| Male | 2216 (42.0) |
| Other/Unknown | 1 (0.0) |
| Race | |
| White | 5876 (53.2) |
| Non-white | 5188 (46.9) |
| Unknown | 21 (0.2) |
| Race with CD | |
| White | 3144 (59.6) |
| Non-white | 2129 (40.3) |
| Unknown | 14 (0.0) |
| BMI (kg/m$^2$) | 29.6 (9.3) |
| Systolic blood pressure (SBP, mmHg) | 124.6 (19.4) |
| Diastolic blood pressure (DBP, mmHg) | 74.4 (14.9) |
| Hemoglobin A1c (%) | 7.11 (1.79) |
| Total cholesterol (mg/dL) | 105.2 (35.9) |
| Current smoking | 2453 (22.2) |
| Number of measures | |
| Total measures | 269475 |
| Maximum measures per patient | 311 |
| Minimum measures per patient | 5 |
| Average measures per patient | 24 |
| Median measures per patient | 17 |
| Cardiac dysrhythmias (CD) | 5271 (47.7) |

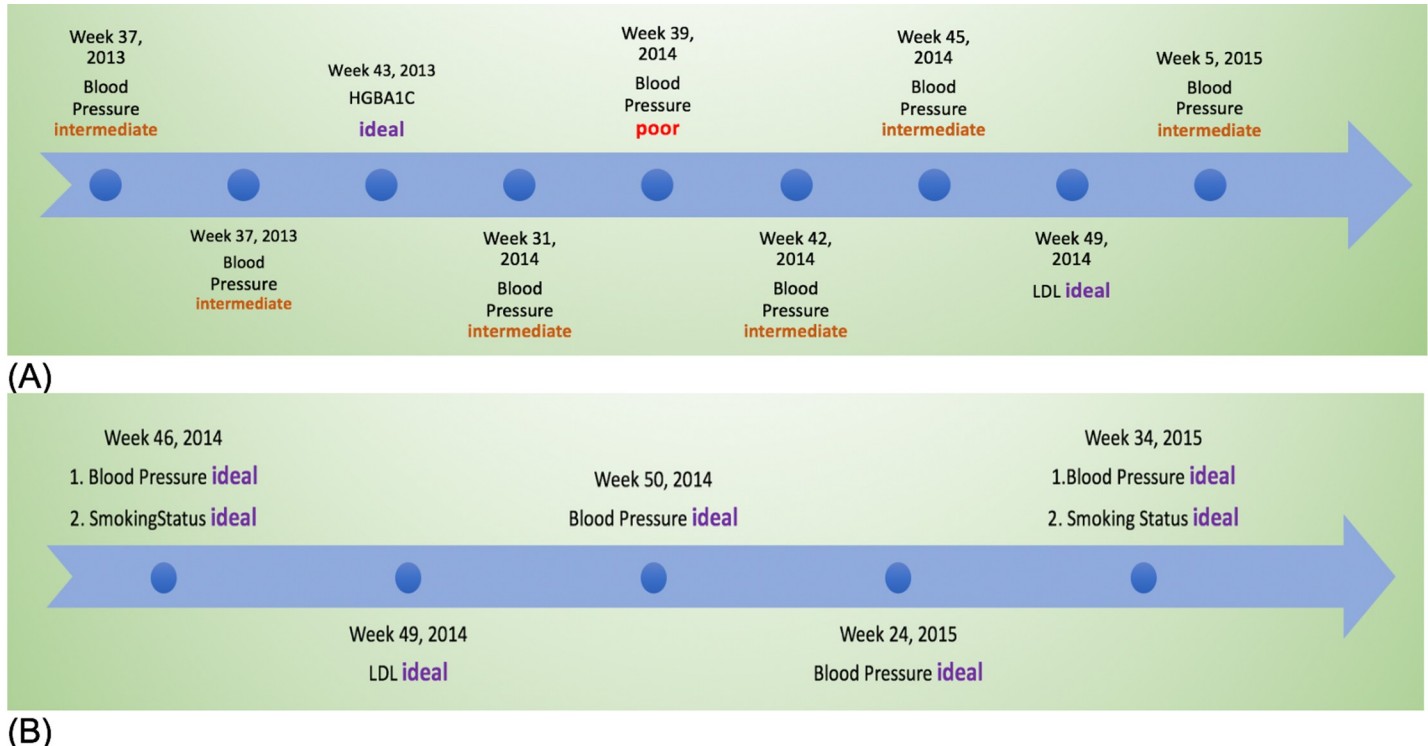

**Fig 2. Examples of CVH time series data.** (A) Patient was diagnosed with CD; (B) Patient not diagnosed with CD.

TSNE is a machine learning technique for visualization by embedding high-dimensional data into a low-dimensional space (here is 2-dimensional space). The features closest to one another in the visualization can be thought of as being more highly correlated with one another.

The LSTM model outperformed other machine learning models in the two cases: inputs with vectors from Word2Vec and inputs with categorical variables (i.e., without Word2Vec).

**Table 3. Characteristics [mean, (SD) or n (%)] of the converted dataset.**

|  | CD = Yes | ideal | intermediate | poor |
|---|---|---|---|---|
| Total unique patients | 5271 |  |  |  |
| Total rows | 128160 | 59315 | 31743 | 37102 |
| Total A1C tests | 6947 | 1073 (15.4) | 2398 (34.5) | 3476 (50.0) |
| Total LDL tests | 11732 | 9310 (79.4) | 1617 (13.8) | 805 (6.9) |
| Total BMI tests | 24532 | 5509 (22.5) | 6870 (28.0) | 12153 (49.5) |
| Total BP tests | 48118 | 15193 (31.6) | 20798 (43.2) | 12127 (25.2) |
| Total Smoking status | 36831 | 28230 (76.6) | 60 (0.2) | 8541 (23.2) |
|  | CD = No | ideal | intermediate | poor |
| Total unique patients | 5784 |  |  |  |
| Total rows | 141315 | 72013 | 32046 | 37256 |
| Total A1C tests | 6065 | 900 (14.8) | 1669 (27.5) | 3496 (57.6) |
| Total LDL tests | 9211 | 6917 (75.1) | 1483 (16.1) | 811 (8.8) |
| Total BMI tests | 31898 | 12358 (38.7) | 6793 (21.3) | 12747 (40.0) |
| Total BP tests | 56532 | 23654 (41.8) | 21745 (38.5) | 11133 (19.7) |
| Total Smoking status | 37609 | 28184 (74.9) | 356 (0.9) | 9069 (24.1) |

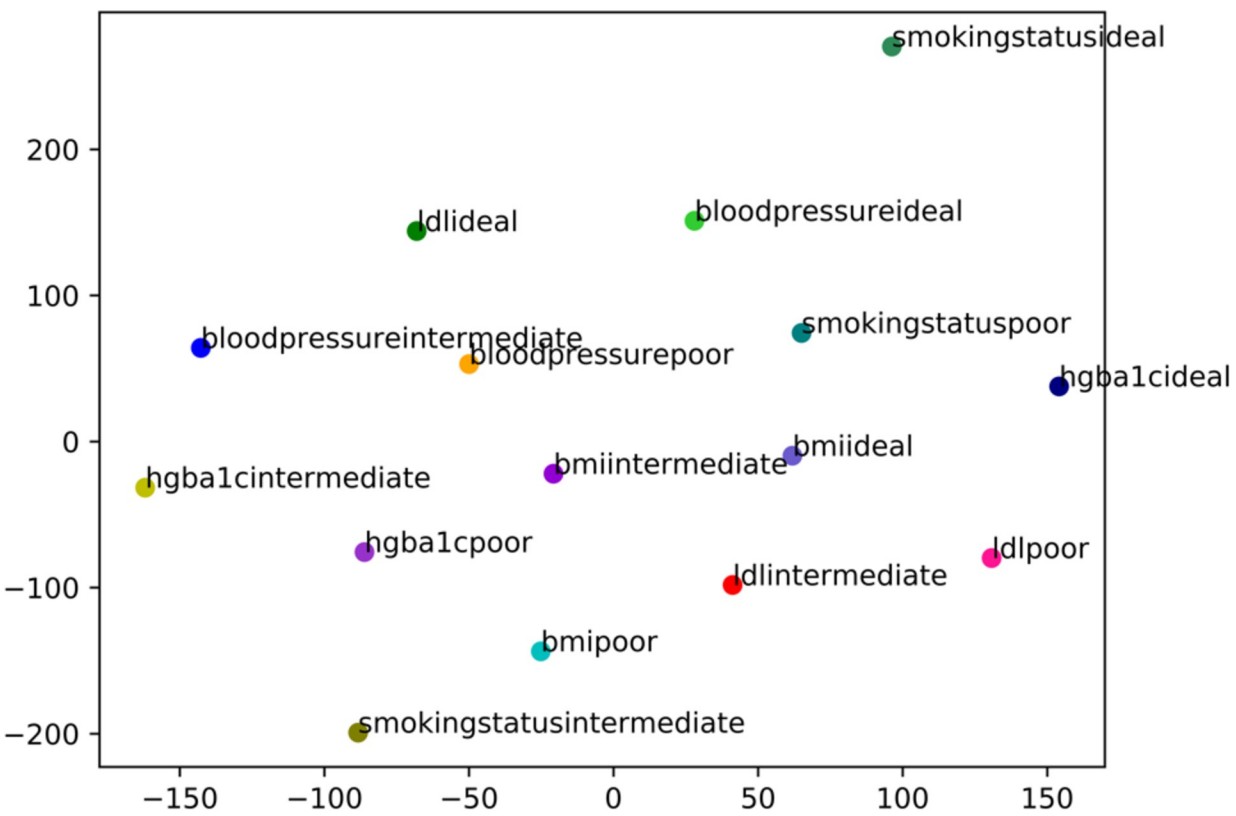

**Fig 3. Embedding visualization of the combination of measure submetric and measure values.** X and y-axes are the first two components in the t-Distributed Stochastic Neighbor Embedding (tSNE).

The AUC of LSTM was 076 (std 0.01) while DNN was 0.71 (std 0.03), LR was 0.67 (std 0.01), RF was 0.66 (std 0.01) and NB was 0.59 (std 0.02) for the case with Word2Vec. For the case without Word2Vec, the AUC of LSTM was 0.69 (std 0.01) while DNN was 0.64 (std 0.02), LR was 0.65 (std 0.01), RF was 0.66 (std 0.01) and NB was 0.60 (std 0.01) (**Fig 4**). The accuracy of each model was 69% for LSTM compared to 66% for DNN, 64% for LR, 61% for RF, and 52% for NB for the case with Word2Vec (**Table 4**). For the case without Word2Vec, the accuracy was 64% for LSTM, 61% for DNN, 62% for RF, 61% for LR, and 52% for NB.

The calculation of metrics was based on the following formulas.

$$Accuracy = (TP + TN)/(TP + TN + FP + FN)$$

$$Sensitivity = TP/(TP + FN)$$

$$Specificity = TN/(TN + FP)$$

$$Precision = TP/(TP + FP)$$

$$F1-score = 2TP/(2TP + FP + FN)$$

Where $TP$ is true positive, $TN$ is true negative, $FP$ is false positive and $FN$ is false negative.

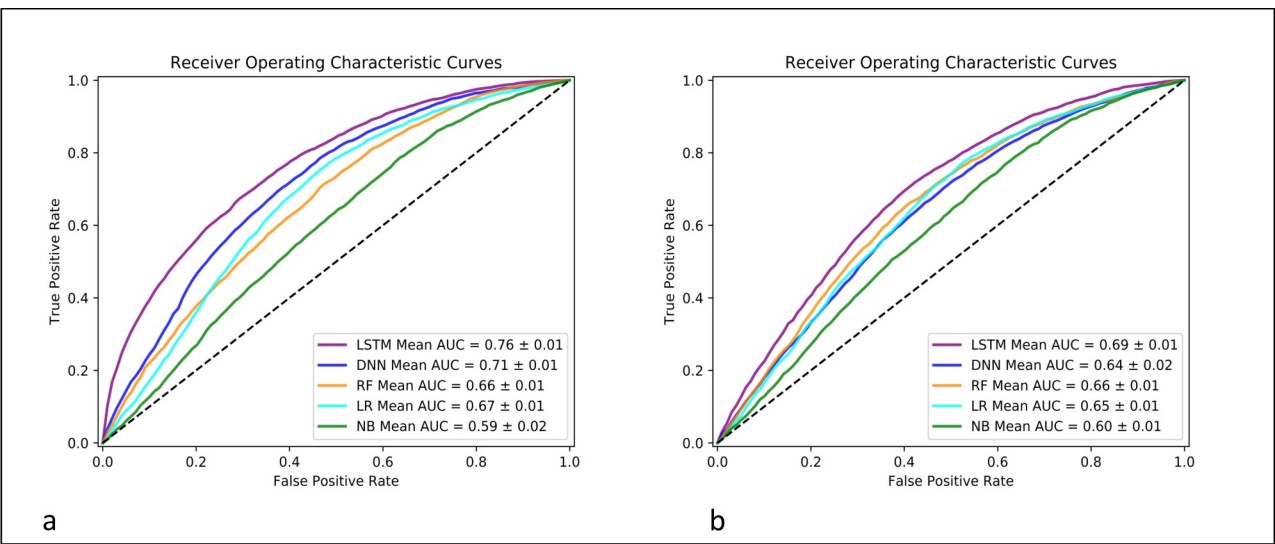

**Fig 4. CD prediction performance by area under the curve (AUC) for LSTM, DNN, RF, LR, and NB models.** LSTM–long short-term memory; RF–random forest; NB–naïve Bayes.

We have also compared the statistical significance for metrics from different models by one-tailed t-test. For example, there were 5 values of accuracy from the 5-fold cross validation for LSTM model and DNN model. We performed a one-tailed t-test on these values of accuracy to determine the statistical significance. The p-values in the **Table 5** show that almost all of the LSTM model performance metrics were significantly higher than other models.

We examined the importance of each feature by evaluating the AUC after removal of the feature from the LSTM model (**Fig 5**). We demonstrated that removing bloodpressureideal and bloodpressureintermediate, the AUC values decreased largely, which indicated that blood pressure contributed to CD prediction largely for LSTM to discriminate CD patients from the healthy group.

## Discussion

In this study, we utilized data from clinics across the US to examine the association between CVH measures and CD diagnoses over a 10-year period by employing traditional machine

**Table 4. Model performance by metrics of 5-fold cross-validation mean (std).**

| Cases | Models | Accuracy | Precision | Recall | f1 | Specificity |
|---|---|---|---|---|---|---|
| **Case: Inputs with vectors by Word2Vec** | LSTM | 0.69 (0.01) | 0.68 (0.02) | 0.66 (0.03) | 0.67 (0.02) | 0.72 (0.03) |
| | DNN | 0.66 (0.03) | 0.63 (0.01) | 0.69 (0.03) | 0.66 (0.01) | 0.63 (0.03) |
| | RF | 0.61 (0.01) | 0.59 (0.01) | 0.61 (0.04) | 0.6 (0.02) | 0.61 (0.02) |
| | LR | 0.64 (0.01) | 0.61 (0.01) | 0.64 (0.01) | 0.63 (0.01) | 0.63 (0.02) |
| | NB | 0.52 (0.0) | 0.0 (0.0) | 0.0 (0.0) | 0.0 (0.0) | 1.0 (0.0) |
| **Case: Inputs without Word2Vec** | LSTM | 0.64 (0.01) | 0.62 (0.02) | 0.65 (0.05) | 0.63 (0.02) | 0.64 (0.04) |
| | DNN | 0.61 (0.01) | 0.59 (0.02) | 0.58 (0.08) | 0.58 (0.04) | 0.63 (0.08) |
| | RF | 0.62 (0.01) | 0.6 (0.01) | 0.61 (0.01) | 0.61 (0.01) | 0.63 (0.02) |
| | LR | 0.61 (0.01) | 0.58 (0.02) | 0.62 (0.01) | 0.6 (0.01) | 0.6 (0.02) |
| | NB | 0.52 (0.0) | 0.6 (0.49) | 0.0 (0.0) | 0.0 (0.01) | 1.0 (0.0) |

**Table 5. Statistical significance of model comparison metrics.**

| Cases | Metrics | (LSTM, DNN) | (LSTM, RF) | (LSTM, LR) | (LSTM, NB) |
|---|---|---|---|---|---|
| **Case: Inputs with vectors by Word2Vec** | **AUC** | $2.2*10^{-4}$ | $3.3*10^{-6}$ | $4.8*10^{-6}$ | $8.0*10^{-8}$ |
| | **Accuracy** | $5.9*10^{-4}$ | $2.9*10^{-5}$ | $2.3*10^{-5}$ | $2.6*10^{-9}$ |
| | **Precision** | $8.8*10^{-4}$ | $3.5*10^{-5}$ | $1.3*10^{-4}$ | $1.4*10^{-12}$ |
| | **Recall** | 0.09 | 0.04 | 0.2 | $3.5*10^{-11}$ |
| | **F1-score** | 0.1 | $5.6*10^{-4}$ | $1.0*10^{-3}$ | $3.3*10^{-13}$ |
| | **Specificity** | 0.003 | $1.5*10^{-4}$ | $3.5*10^{-4}$ | $5.3*10^{-8}$ |
| **Case: Inputs without Word2Vec** | **AUC** | $8.0*10^{-4}$ | $2.3*10^{-4}$ | $1.7*10^{-4}$ | $5.7*10^{-8}$ |
| | **Accuracy** | 0.04 | 0.02 | 0.04 | $6.3*10^{-8}$ |
| | **Precision** | 0.04 | 0.08 | 0.006 | 0.46 |
| | **Recall** | 0.1 | 0.08 | 0.1 | $2.0*10^{-9}$ |
| | **F1-score** | 0.02 | 0.01 | 0.06 | $1.6*10^{-12}$ |
| | **Specificity** | 0.42 | 0.44 | 0.09 | $1.0*10^{-7}$ |

learning models and deep learning techniques. Our results indicated that the deep learning algorithm–LSTM–outperformed other traditional machine learning algorithms and achieved the best accuracy performance. Our results in which we dropped one feature at a time by setting it as constant from our analyses indicated that patients with ideal CVH especially blood pressure was associated with diagnosis with CD, which was consistent with results that high blood pressure and obesity might increase the chances of developing a CD. Additional risk factors that were associated with a higher prevalence of CD included female gender and white race. Our study is the first to utilize the LSTM to investigate the relationship between

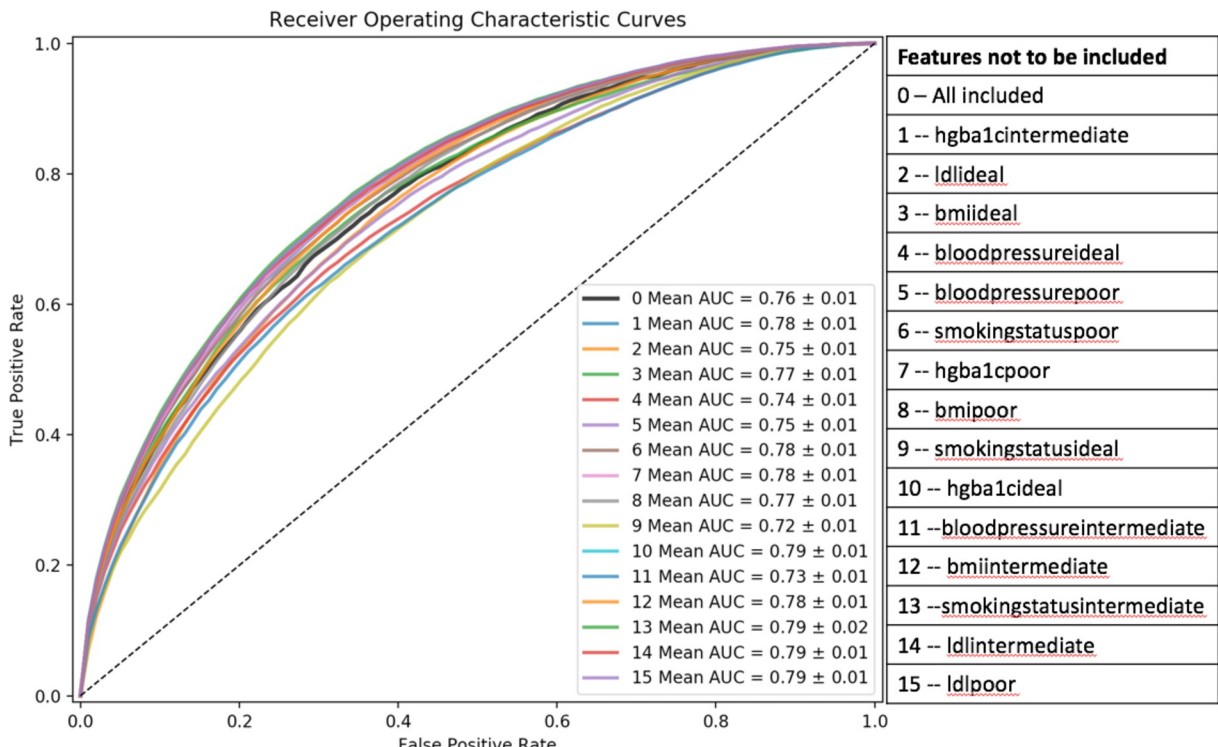

**Fig 5. Feature discriminative importance evaluated using the LSTM model.**

time-series CVH measurements and CD diagnoses. Not surprisingly, the LSTM deep learning model achieved the best performance compared with the traditional machine learning algorithms used in previous EHR data studies [34]. An advantage of our study is that the results represent associations seen in over 70 clinics in the US. In addition, we will investigate more about the time-aware LSTM models [35, 36] to better capture the underlying patterns in the irregular time intervals in the longitudinal EHR data.

## Limitations

We encountered some limitations to using EHR data for these analyses. First, patients had different times for visits of CVH measurements as some patients visited more frequently and had high numbers of visits and some just had a few visits. To address this, we created virtual events for patients with fewer visits in order to conduct our analyses. Second, the prediction accuracy might be further improved with additional demographic and clinical factors in addition to the regular medical visits and measurements used in this study (e.g., health data collected from wearable devices). Our findings would be much more generalizable if we had greater representation from more clinics across the U.S.

## Conclusions

Deep learning models can effectively predict incident CD from time-series CVH measurements compared with traditional machine learning algorithms. Ideal CVH scores, especially BMI and blood pressure, could be associated with lower chance of developing CD. This study determined the extent to which ideal CVH is important to attain and maintain for more favorable outcomes. These findings may be used to prevent CD in the outpatient setting by encouraging appropriate management of CVH.

## Author Contributions

**Conceptualization:** Randi E. Foraker.

**Formal analysis:** Aixia Guo.

**Supervision:** Randi E. Foraker.

**Writing – original draft:** Aixia Guo.

**Writing – review & editing:** Sakima Smith, Yosef M. Khan, James R. Langabeer II, Randi E. Foraker.

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
