## [Decision Letter · Decision Letter 0]

9 Jun 2020

PONE-D-20-10776

Application of a time-series deep learning model to predict cardiac dysrhythmias in electronic health records

PLOS ONE

Dear Dr. Foraker,

Thank you for submitting your manuscript to PLOS ONE. After careful consideration, we feel that it has merit but does not fully meet PLOS ONE’s publication criteria as it currently stands. Therefore, we invite you to submit a revised version of the manuscript that addresses the points raised during the review process.

We look forward to receiving your revised manuscript.

Kind regards,

Sreeram V. Ramagopalan

Academic Editor

PLOS ONE

Journal Requirements:

2. Our internal editors have evaluated your manuscript and determined that it is within the scope of our 'Primary and Secondary Prevention of Cardiovascular Disease' Call for Papers. This collection of papers is headed by a team of Guest Editors for PLOS ONE and will encompass a diverse range of research articles. Additional information can be found on our announcement page: (https://collections.plos.org/s/prevention-cardiovascular). If you would like your manuscript to be considered for this collection, please let us know in your cover letter and we will ensure that your paper is treated as if you were responding to this call. If you would prefer to remove your manuscript from collection consideration, please specify this in the cover letter.

3. In ethics statement in the manuscript and in the online submission form, please provide additional information about the patient records used in your retrospective study. Specifically, please ensure that you have discussed whether all data were fully anonymized before you accessed them and/or whether the IRB or ethics committee waived the requirement for informed consent. If patients provided informed written consent to have data from their medical records used in research, please include this information.

Additional Editor Comments (if provided):

Reviewers' comments:

Reviewer's Responses to Questions

**Comments to the Author**

1. Is the manuscript technically sound, and do the data support the conclusions?

Reviewer #1: No

2. Has the statistical analysis been performed appropriately and rigorously? 

Reviewer #1: No

3. Have the authors made all data underlying the findings in their manuscript fully available?

Reviewer #1: No

4. Is the manuscript presented in an intelligible fashion and written in standard English?

Reviewer #1: Yes

5. Review Comments to the Author

Reviewer #1: In their manuscript titled "Application of a time-series deep learning model to predict cardiac dysrhythmias in electronic health records", Guo et al use the Electronic healthcare records (EHR) data available in The Guideline Advantage (TGA) and deep learning methodologies to make predictions on cardiac dysrythimas (CD) based on cardiovascular health (CVH) measures. They propose that LSTM performs better than LR, RF and NB and identify blood pressure and BMI as the most influential features.

Extracting time series information based on EHR data is a topic of high interest in the community and the question the authors pose, namely "can we predict CD based on EHR data using machine learning methodologies?" is very timely because the quality and quantity of available data is increasing and machine learning techniques have become not only more powerful but also more interpretable and robust. The authors states that the paper is the first time that DL algorithms have been applied to predict CD using time series EHR data. However, the manuscript has two major shortcomings; the methodology is not described in detail to be reproducible and the time dependent information does not reflect real time.

Time dependence: DL has been used for different classification tasks on EHR and CD prediction using different types of datasets (Solares 2020, Xia 2018, Shashikumar 2017). This study builds on literature by applying existing methods to a previously studied dataset. The authors state that this is the first that DL is applied to predict CD using time series EHR data. Indeed, if time dependence is incorporated into the predictive model, this is very interesting. However, it is not clear how time is included in the representations. The authors also mention this in the Limitations section. The events are represented by 311 time steps. Are two events separated by two days treated the same way as two events separated by two months? The authors state they use padding to create events so that they can have a 311D vector. How is this padding performed? If dynamic data is the major contribution of the paper, more information about the treatment of time should be included. Furthermore, it is not clear how blood pressure and BMI are the most influential features are impacted by time dependence.

Word2Vec: The study first constructs categorical features to represent patients and then learns continuous vectors usingWord2Vec algorithm. The authors can compare Word2Vec with categorical representations to show the value added by distributed representations. Besides, the construction of categorical vectors and the association of vectors with time can be detailed further, since these are key to the representations.

LSTM: LSTM is adopted as a neural time-series model to identify CD patients and it is compared with three benchmark models: Naive Bayes, Logistic Regression and Random Forest. Given that these models are neither designed to classify time-series nor as complex as a neural network, other models, such as non-neural time-series models and feed-forward neural networks, can be added as benchmarks to reveal the benefits of LSTM further. This would also uncover if the performance gain is due to time-series representation or complexity of neural models.

Causation: The study concludes that “BMI and blood pressure could lower the chance of developing CD” based on the fact that the removal of these features affected the prediction performance the most. This association can be analyzed in more detail, since these features are first embedded using Word2Vec and, thus, not directly observed by the prediction model. Moreover, the causation can be shown explicitly, since the performance drop might not necessarily indicate a causation but might simply mean that these features helped model to discriminate CD patients from the healthy group the most.

DL model: The code is not yet available. The authors have not included important details about the algorithms used. For example, how do the authors test for overfit? Is 5-fold cross validation performed? What is the error in their performance metrics? What is the statistical significance of their model comparison metrics?

The manuscript is well organized and easy to follow but more details about the methodology are needed and the conclusions are not supported by the experiments.

6. PLOS authors have the option to publish the peer review history of their article (what does this mean?). If published, this will include your full peer review and any attached files.

Reviewer #1: No

---

## [Author Response · Author response to Decision Letter 0]

24 Jul 2020

We would like to thank the editor and the reviewer for the great and helpful comments. Please find our point-by-point response to the reviews and editors in the attached file 'Response to reviewers'.

---

## [Decision Letter · Decision Letter 1]

13 Oct 2020

PONE-D-20-10776R1

Application of a time-series deep learning model to predict cardiac dysrhythmias in electronic health records

PLOS ONE

Dear Dr. Foraker,

Thank you for submitting your manuscript to PLOS ONE. After careful consideration, we feel that it has merit but does not fully meet PLOS ONE’s publication criteria as it currently stands. Therefore, we invite you to submit a revised version of the manuscript that addresses the points raised during the review process.

Dear Authors, you have since the acceptance of this manuscript, resubmitted the manuscript with changes stating that analyses were revised and different results obtained. On the version that I received, these changes were not reflected in the abstract and elsewhere in the manuscript. As these changes were materially impacting the manuscript we will send the manuscript for re-review. If you do wish to resubmit the manuscript please make sure the updated manuscript fully accounts for the new results, otherwise we will not be willing to proceed further with you manuscript.

We look forward to receiving your revised manuscript.

Kind regards,

Sreeram V. Ramagopalan

Academic Editor

PLOS ONE

Reviewers' comments:

Reviewer's Responses to Questions

**Comments to the Author**

1. If the authors have adequately addressed your comments raised in a previous round of review and you feel that this manuscript is now acceptable for publication, you may indicate that here to bypass the “Comments to the Author” section, enter your conflict of interest statement in the “Confidential to Editor” section, and submit your "Accept" recommendation.

Reviewer #1: All comments have been addressed

2. Is the manuscript technically sound, and do the data support the conclusions?

Reviewer #1: Yes

3. Has the statistical analysis been performed appropriately and rigorously? 

Reviewer #1: Yes

4. Have the authors made all data underlying the findings in their manuscript fully available?

Reviewer #1: Yes

5. Is the manuscript presented in an intelligible fashion and written in standard English?

Reviewer #1: Yes

6. Review Comments to the Author

Reviewer #1: The authors have addressed the concerns. The authors can include some references to time aware LSTM for future work (eg DOI: 10.1145/3097983.3097997).

7. PLOS authors have the option to publish the peer review history of their article (what does this mean?). If published, this will include your full peer review and any attached files.

Reviewer #1: No

---

## [Author Response · Author response to Decision Letter 1]

21 Oct 2020

Response to reviewer comments

We would like to thank the reviewers for their informed, thoughtful, and helpful comments. Please find our responses to the reviews below in italics. We believe that the manuscript has been significantly improved by our responses to the reviewers and hope that you will find it suitable for publication in the PLOS ONE.

Reviewers' comments:

Reviewer's Responses to Questions

Comments to the Author

1. If the authors have adequately addressed your comments raised in a previous round of review and you feel that this manuscript is now acceptable for publication, you may indicate that here to bypass the “Comments to the Author” section, enter your conflict of interest statement in the “Confidential to Editor” section, and submit your "Accept" recommendation.

Reviewer #1: All comments have been addressed

2. Is the manuscript technically sound, and do the data support the conclusions?

Reviewer #1: Yes

3. Has the statistical analysis been performed appropriately and rigorously? 

Reviewer #1: Yes

4. Have the authors made all data underlying the findings in their manuscript fully available?

Reviewer #1: Yes

5. Is the manuscript presented in an intelligible fashion and written in standard English?

Reviewer #1: Yes

6. Review Comments to the Author

Reviewer #1: 

The authors have addressed the concerns. The authors can include some references to time aware LSTM for future work (eg DOI: 10.1145/3097983.3097997).

Thank you. Our apologies for not adding these references in the original manuscript. We have added the following references and also added to the discussion section as the future work.

1.Baytas IM, Xiao C, Zhang X, Wang F, Jain AK, Zhou J. Patient subtyping via time-aware LSTM networks. In: Proceedings of the ACM SIGKDD International Conference on Knowledge Discovery and Data Mining. 2017.

2.Zhang Y, Yang X, Ivy J, Chi M. Attain: Attention-based time-aware LSTM networks for disease progression modeling. IJCAI International Joint Conference on Artificial Intelligence. 2019. doi:10.24963/ijcai.2019/607.

Dear reviewer,

We made an error in data pre-processing during the last revision process. We have since found the error, corrected it, and reproduced the related results (specifically, Figures 4 & 5, and Tables 4 & 5). All of the original conclusions and patterns still hold and the values (Figures 4 & 5) of the results are similar to the initial submission, prior to the submitted revision. Tables 4 & 5 were newly added during the last revision by suggestions from reviewers.

To provide additional context, the final values of the results (Figures 4 & 5, and Tables 4 & 5) we now include in the manuscript for your kind consideration.

We apologize for any inconvenience this may cause and are grateful for your time and expertise in assessing our manuscript. We hope you find the revised manuscript suitable for publication; in the meantime, please let us know if you have any questions or concerns.

Sincerely,

Randi Foraker, PhD, MA, FAHA

Associate Professor of Medicine

Washington university in St Louis

---

## [Editor Report · Decision Letter 2]

6 Nov 2020

Application of a time-series deep learning model to predict cardiac dysrhythmias in electronic health records

PONE-D-20-10776R2

Dear Dr. Foraker,

We’re pleased to inform you that your manuscript has been judged scientifically suitable for publication and will be formally accepted for publication once it meets all outstanding technical requirements.

Kind regards,

Sreeram V. Ramagopalan

Academic Editor

PLOS ONE
---

## [Editor Report · Acceptance letter]

20 Nov 2020

PONE-D-20-10776R2 

Application of a time-series deep learning model to predict cardiac dysrhythmias in electronic health records 

Dear Dr. Foraker:

I'm pleased to inform you that your manuscript has been deemed suitable for publication in PLOS ONE. Congratulations! Your manuscript is now with our production department. 

Kind regards, 

on behalf of

Dr. Sreeram V. Ramagopalan 

Academic Editor

PLOS ONE